# Beraprost Sodium for Pulmonary Hypertension in Dogs: Effect on Hemodynamics and Cardiac Function

**DOI:** 10.3390/ani12162078

**Published:** 2022-08-15

**Authors:** Ryohei Suzuki, Yunosuke Yuchi, Takahiro Saito, Yuyo Yasumura, Takahiro Teshima, Hirotaka Matsumoto, Hidekazu Koyama

**Affiliations:** Laboratory of Veterinary Internal Medicine, School of Veterinary Science, Faculty of Veterinary Medicine, Nippon Veterinary and Life Science University, Tokyo 180-8602, Japan

**Keywords:** dog, impedance, myocardial strain, post-capillary, pre-capillary, prostacyclin, pulmonary circulation, pulmonary vasodilator, speckle-tracking echocardiography, systemic circulation

## Abstract

**Simple Summary:**

Pulmonary hypertension is a potentially life-threatening disease among dogs that is characterized by increased pulmonary arterial pressure and pulmonary vascular resistance. In veterinary medicine, a phosphodiesterase-5 inhibitor such as sildenafil is the most common drug used to treat pulmonary hypertension. However, the availability of sildenafil is limited because of its high cost, difficulty in obtaining the drug in some areas, and potential inter-individual variability in the response to sildenafil therapy. Beraprost sodium is one of the most common drugs used to treat pulmonary hypertension in humans. However, little is known about its efficacy in dogs with pulmonary hypertension. In this study, beraprost sodium showed significant pulmonary and systemic vasodilation without any adverse effects in sixteen dogs with pulmonary hypertension. Additionally, echocardiographic improvements in cardiac function and pulmonary and systemic circulation were observed. These results emphasize the potential efficacy of beraprost sodium in treating canine pulmonary hypertension.

**Abstract:**

Pulmonary hypertension (PH) is a fatal condition that affects many dogs. In humans, PH is often treated with beraprost sodium (BPS). However, the effectiveness of BPS for canine PH has not been established. This study aimed to evaluate the clinical and cardiovascular response of BPS in canine patients with PH of various causes. Sixteen dogs with PH (post-capillary PH, *n* = 8; pre-capillary PH, *n* = 8) were included. BPS was continuously administered twice daily at 15 µg/kg. All dogs underwent echocardiography, including speckle-tracking analysis and blood pressure measurement, before and after BPS administration. Continuous BPS administration (range: 13.2–22.0 µg/kg) significantly decreased the pulmonary and systemic vascular impedance and increased left and right ventricular myocardial strain. In dogs with post-capillary PH, BPS administration caused no significant worsening of the left atrial pressure indicators. No side effects of BPS were observed in any dog. BPS also improved cardiac function and pulmonary circulation through pulmonary vasodilation, suggesting that BPS may be an additional treatment option for canine PH of various causes. Particularly, BPS increased left ventricular function and systemic circulation without worsening the left heart loading condition in dogs with post-capillary PH.

## 1. Introduction

Recent studies on canines have reported that pulmonary hypertension (PH) is a life-threatening complication in dogs with various cardiac and respiratory diseases [1,2,3]. In veterinary medicine, based on its causal factor, PH is hemodynamically classified into two subtypes: pre- and post-capillary PH [1]; both causes of PH can lead to right ventricular (RV) dysfunction, RV enlargement, and right heart failure (RHF) owing to increased RV pressure overload [1,4].

Pulmonary vasodilators are commonly used to treat PH, and phosphodiesterase-5 inhibitors such as sildenafil are the most common drugs used to treat canine PH [1,5,6]. However, the availability of sildenafil is limited because of its high cost, difficulty in obtaining the drug in some areas, and potential inter-individual variability in the response to sildenafil therapy due to genetic polymorphism [7]. Furthermore, in dogs and human patients with post-capillary PH, sildenafil-induced increased pulmonary circulation may elevate the already high left atrial pressure with the consequent risk of precipitating left congestive heart failure or worsening an ongoing lung edema [1,8,9,10]. In humans, beraprost sodium (BPS), a chemically stable prostaglandin I_2_ analog, is reportedly effective against PH because of its vasodilating effects on pulmonary and systemic arterial smooth muscle cells, protective effects on vascular endothelial cells, inhibition of inflammatory cytokine production, and antiplatelet effects [11,12,13,14]. We previously reported that BPS decreased pulmonary and systemic vascular impedance (PVI and SVI, respectively) and increased pulmonary and systemic circulation in experimentally induced canine models of chronic embolic PH [15]. However, no study has investigated the efficacy of BPS in canine patients with clinically diagnosed PH of various causes.

Thus, this study aimed to investigate the clinical and cardiovascular response of BPS in dogs with PH of various etiologies. We hypothesized that BPS would decrease PVI and SVI and increase cardiac function and pulmonary and systemic circulation, like our previous study on canine models of chronic PH.

## 2. Materials and Methods

In this prospective cohort study, client-owned dogs that underwent cardiac examination at Nippon Veterinary and Life Science University Veterinary Medical Teaching Hospital were enrolled from April 2019 to May 2022. All study protocols followed the Guidelines for Institutional Laboratory Animal Care and Use of Nippon Veterinary and Life Science University in Japan, and the study was approved by the Ethical Committee for Animal Use of Nippon Veterinary and Life Science University Veterinary Medical Teaching Hospital, Japan (approval number: R2–5). All dog owners signed a written informed consent form to authorize their participation in this study.

### 2.1. Patients

This study prospectively included client-owned dogs that were clinically diagnosed with PH. All dogs underwent a complete physical examination, oscillometric method-derived blood pressure measurement (BP100DII, FUKUDA M-E KOGYO Co, LTD, Tokyo, Japan), 6-leads electrocardiography, and radiographic and echocardiographic examinations. Electrocardiography was performed for 30 s at least twice. We clinically diagnosed PH based on a high probability of PH noted in the American College of Veterinary Internal Medicine (ACVIM) consensus statement, which was defined using echocardiographic findings of tricuspid regurgitation (TR) velocity (>3.4 m/s) and anatomical abnormalities of the right heart, pulmonary artery, and caudal vena cava [1]. Additionally, dogs were classified into one of two PH subgroups according to the diagnostic and classification criteria for PH described in the ACVIM consensus statement: post- and pre-capillary PH [1]. Furthermore, dogs were determined to have RHF if they exhibited at least one clinical evidence of ascites, pleural effusion, and/or pericardial effusion without any abnormalities other than PH that might have been responsible [15,16]. We also questioned the dogs’ owner about the clinical signs suggestive of PH, such as syncope, respiratory distress at rest, and exercise intolerance [1]. In this study, the severity of PH was estimated according to the TR velocity [16,17,18]. Mild PH was identified if dogs had a TR velocity < 3.5 m/s without any evidence of RHF; moderate PH was identified if dogs had a TR velocity of 3.5–4.3 m/s without any evidence of RHF; severe PH was identified if dogs had a TR velocity > 4.3 m/s [16,17,18]. Additionally, dogs with RHF were considered as having severe PH regardless of the TR velocity.

### 2.2. Study Protocol

All dogs were orally administered BPS twice daily using 55-µg tablets (Toray Industries, Inc., Tokyo, Japan), divided into doses of approximately 15 µg/kg. The dosage and frequency of BPS were determined according to an experimental study on a canine model of PH and a clinical study on cats with chronic kidney disease [15,19]. BPS administration was continued for more than 1 week. In all dogs with PH, BPS was prescribed in addition to the treatment for the causative disease. No adjustments or changes to oral medications other than BPS were made. If any adverse effects were observed owing to BPS (e.g., shock, hemostatic abnormality, and hypotension), its administration was immediately discontinued.

Before and after BPS administration, all dogs underwent complete physical examination, transthoracic radiography, echocardiography, and oscillometric method-derived blood pressure measurements. All examinations were performed without sedation.

### 2.3. Echocardiography

Two-dimensional and Doppler echocardiographic images were obtained by a single investigator (R.S.) using a Vivid E95 Ultra Edition (GE Healthcare, Tokyo, Japan) with a 3.5–6.9 MHz transducer and simultaneous lead II electrocardiography recording. Dogs without sedation were manually restrained in left and right lateral recumbency. All echocardiographic measurements were performed by another observer (Y.Yu.) who had been trained by an operator with years of experience in echocardiography using an offline workstation (EchoPAC PC, version 204; GE Healthcare, Tokyo, Japan). All echocardiographic variables were obtained from three consecutive cardiac cycles, and the average values were used for statistical analyses.

The left atrial-to-aortic-root ratio (LA/Ao); left ventricular (LV) internal dimensions at end-diastole and end-systole normalized by body weight (LVIDDN and LVIDSN, respectively) [20]; and biplane modified Simpson’s method-derived LV volumes at end-diastole and end-systole normalized by body surface area (LVEDVI and LVESVI, respectively) were obtained as the left heart morphological indicators [20,21,22,23,24]. Additionally, LV fractional shortening and ejection fraction were calculated using the LV internal dimension and LV volume, respectively [23]. Depending on the availability, mitral regurgitation (MR) velocity was determined using the continuous-wave spectral Doppler method. Transmitral inflow was obtained using the left apical four-chamber view and Doppler echocardiography, and early diastolic and late diastolic wave velocities (E and A, respectively) were measured [25]. Tissue Doppler imaging-derived peak myocardial velocity of the septal mitral annulus at early-diastole (e’) was also measured [26]. Using these Doppler echocardiographic variables, E/A and E/e’ were calculated [27].

To assess RV morphology, RV areas at end-diastole and end-systole normalized by body weight (RVEDA and RVESA indices, respectively) were obtained. Additionally, RV fractional area change (FAC), tricuspid annular plane systolic excursion (TAPSE) obtained using the B-mode method, and tissue Doppler imaging-derived peak myocardial velocity of the lateral tricuspid annulus at systole (RV s’) were measured as RV functional indicators. The left apical four-chamber view, optimized for the right heart (RV focus view), was used for all RV morphological and functional variables [15,28,29,30]. RV FAC and TAPSE were normalized by body weight using the following formulas (RV FACn and TAPSEn, respectively) [15,28,31]: RV FACn = (RV FAC [%])/(body weight [kg])^−0.097^(1)
TAPSEn = (TAPSE [mm])/(body weight [kg])^0.284^(2)

The PVI and SVI were calculated as indicators of pulmonary and systemic vascular resistance, respectively [15]. SVI was calculated using the following formula employing the oscillometric method-derived systolic systemic arterial pressure and LV stroke volume normalized by the body surface area (LV SV) [15]:SVI = (systemic arterial pressure [mmHg])/(LV SV [mL/m^2^])(3)

Additionally, PVI was calculated using the following formula employing the TR pressure gradient and RV stroke volume normalized by the body surface area (RV SV) [15]:PVI = 4 × (TR velocity [m/s])^2^/(RV SV [mL/m^2^])(4)

LV and RV stroke volumes were estimated using the cross-sectional area method as previously described [32]. TR was identified by inspecting the tricuspid valve from multiple views and using color Doppler echocardiography, and peak velocity was measured using the continuous-wave spectral Doppler method.

Two-dimensional speckle-tracking echocardiography (2D-STE) was performed as the indicator of the precise LV and RV function. LV longitudinal and circumferential strains (LV-SL and LV-SC, respectively) were measured using the left apical four-chamber view and right parasternal short-axis view at the level of the papillary muscle [15,33,34]. The RV longitudinal strain (RV-SL) was measured as a precise RV functional indicator using the RV focus view [15,16,35,36]. RV-SL was measured using the LV four-chamber algorithm. In this study, only the RV free wall was used to measure RV-SL. The analysis procedure of 2D-STE has been described previously [15,16,33,34,36,37,38]. All strains were defined as the absolute values of the negative peaks from each strain wave [15,33,34,37].

### 2.4. Statistical Analysis

Commercially available software was used for all statistical analyses (EZR version 1.41, Saitama Medical Center, Jichi Medical University, Saitama, Japan) [34]. All continuous variables are expressed as median (interquartile range). Normality of the data was evaluated using the Shapiro–Wilk test. Paired *t*-tests (for normally distributed data) or Wilcoxon signed-rank sum tests (for non-normally distributed data) were used for changes in continuous variables. Pre- and post-examination values were compared separately for dogs with post- and pre-capillary PH. *p*-values < 0.050 were considered statistically significant.

## 3. Results

### 3.1. Clinical Characteristics

All the dogs with PH completed the study protocol. Table 1 shows the clinical characteristics of 16 dogs with PH before and after BPS administration. The causes of pre-capillary PH were hypoxia due to various respiratory diseases (*n* = 5), reversed patent duct arteriosus (*n* = 1), filariasis (*n* = 1), and chronic pulmonary thromboembolic disease attributed to hyperadrenocorticism (*n* = 1). All cases of post-capillary PH were caused by myxomatous mitral valve disease (ACVIM stage: one dog was identified as stage B2 and the others were stage C or D). All dogs with PH received some medications for PH or causative diseases of PH at the pre-examination (Table 2). There were no changes in medications other than BPS during the study period in all dogs with PH. A median dose of 15.9 µg/kg of BPS (range: 13.4–22.0 µg/kg) was continuously administered for at least 1 week (range: 7–21 days). Supraventricular premature contraction was observed in three dogs with post-capillary PH and disappeared in one dog after BPS administration. Ten dogs with PH (five each with pre- and post-capillary PH) showed clinical evidence of RHF. The clinical findings indicating the RHF observed before the BPS administration in dogs with pre- and post-capillary PH were as follows: pre-capillary PH, ascites (*n* = 2), ascites and pleural effusion (*n* = 1); post-capillary PH, ascites (*n* = 2), ascites and pericardial effusion (*n* = 2), pericardial effusion (*n*= 1). All dogs had at least one clinical sign suggestive of PH at the pre-examination. Post BPS administration, RHF disappeared in three dogs with post-capillary PH and in one with pre-capillary PH. Two dogs with severe PH were improved to moderate PH after BPS administration. Additionally, some dogs showed improvement in clinical signs suggestive of PH, including syncope, weakness, and exercise intolerance. No side effects associated with BPS administration were observed in any dog with PH.

### 3.2. Echocardiographic Variables

Table 3 and Figure 1 show the echocardiographic variables for the left heart morphology and function before and after BPS administration in 16 dogs with PH. In dogs with post-capillary PH, there were no significant changes in the left heart morphological indicators. Left atrial pressure indicators, such as E, E/A, and E/e’, exhibited no significant changes (*p* = 0.354, 0.362, and 0.230, respectively), and MR velocity significantly increased with BPS administration (*p* = 0.024). BPS administration also significantly decreased the SVI (*p* = 0.012) (Figure 1b). Although there was no significant change in fractional shortening (*p* = 0.579), ejection fraction, LV SV (Figure 1a), LV-SL (Figure 1c), and LV-SC (Figure 1d) significantly increased with BPS administration (*p* = 0.004, 0.012, and 0.002, respectively). Conversely, in dogs with pre-capillary PH, LVEDVI, ejection fraction, LV SV (Figure 1a), LV-SL (Figure 1c), and LV-SC (Figure 1d) significantly increased after BPS administration (*p* = 0.013, 0.002, <0.001, and 0.013, respectively). Additionally, BPS administration significantly decreased the SVI (*p* = 0.008) (Figure 1b).

Echocardiographic variables for the right heart morphology and function are summarized in Table 4 and Figure 2. In dogs with post-capillary PH, RV FACn, RV SV (Figure 2a), and RV-SL (Figure 2c) significantly increased following BPS administration (*p* = 0.006, 0.012, and <0.001, respectively). Additionally, BPS administration significantly decreased PVI (Figure 2b) and TR velocity (*p* < 0.001 and *p* = 0.033, respectively). The same results of RV FACn, RV SV (Figure 2a), RV-SL (Figure 2c), PVI (Figure 2b), and TR velocity were observed in dogs with pre-capillary PH (*p* = 0.034, 0.031, 0.001, 0.004, and <0.001, respectively). However, increased RV size indicators (RVEDA and RVESA) showed no significant improvement with BPS administration (*p* = 0.221 and *p* = 0.051, respectively).

## 4. Discussion

In this study, BPS showed clinical and echocardiographic improvements in dogs with PH. Significantly decreased PVI and TR velocities were observed in dogs with PH via its pulmonary vasodilating effect. Additionally, the decrease in RV pressure overload significantly improved RV function and pulmonary circulation in dogs with PH. These results suggest that BPS may be a treatment option for canine patients with PH, regardless of the causative diseases. Furthermore, in dogs with post-capillary PH, LV function and systemic circulation significantly improved without significant increases in left heart size or left atrial pressure indicators. Our results also indicate that BPS may improve both pulmonic and systemic circulation without left heart loading in dogs with post-capillary PH.

BPS significantly improved RV hemodynamics and loading conditions based on the decrease in PVI and TR velocity. Additionally, a significant increase in pulmonary circulation and RV functional indicators, including RV-SL, was observed after BPS administration. These results are consistent with those of a previous study in which BPS decreased RV pressure overload and increased RV function in a dose-dependent manner in a canine model of chronic embolic PH [15]. Therefore, 15 µg/kg BPS may be an additional treatment option for patients with PH. Additionally, although we did not evaluate the antiplatelet effect of BPS, it may also be effective, especially in dogs with PH with a potential for causing pulmonary thromboembolism. However, it is important to consider that further investigations are needed to properly characterize the antiplatelet action of this molecule in naturally acquired thromboembolic diseases in dogs.

In human patients with post-capillary PH, improved pulmonary circulation may burden the left heart and increase elevated left atrial pressure [8]. Previous studies have reported adverse effects of sildenafil, which elevated left atrial pressure and induced pulmonary edema in patients with PH [8,9,10]. In this study, however, there were no significant changes in the left heart size indicators (such as LA/Ao, LVIDDN, and LVEDVI). Furthermore, no significant worsening was observed in the left atrial pressure indicators (E, E/A, E/e’, and MR velocity) after BPS administration. The main cause of these results was the systemic vasodilating effect of BPS. In our previous study on canine models of chronic embolic PH, a BPS of 15 µg/kg showed a balanced vasodilatory effect in both pulmonary and systemic vessels [15]. In this study, both the PVI and SVI significantly decreased with BPS administration. Additionally, significant increases in ejection fraction, LV SV, and speckle-tracking echocardiography-derived LV-SL and LV-SC were observed. These results suggest that BPS could be effective in treating post-capillary PH without worsening the left heart load through both pulmonic and systemic vasodilating effects and subsequent improvements in RV and LV function. Additionally, improved LV systolic function might also influence the increase in MR velocity.

In dogs with pre-capillary PH, BPS administration significantly improved PVI, TR velocity, and RV functional indicators including RV FACn, RV SV, and RV-SL. Additionally, a significant improvement in LV functional indicators such as ejection fraction, LV SV, and LV-SL was observed. The improved LV function in dogs with pre-capillary PH was possibly due to increased LV filling (i.e., increased LVEDVI), in agreement with the Frank–Starling law, in addition to the systemic vasodilating effect of BPS, which reduced the LV afterload, thus favouring the systolic antegrade flow. Furthermore, the antiplatelet and anti-inflammatory effects of BPS may also contribute to the reduction in PVI and consequently improve pulmonary circulation and LV filling [12,13,39]. Overall, these results support BPS effectiveness for treating pre-capillary PH. However, BPS administration did not significantly decrease the enlarged RV size in dogs with pre-capillary PH. Furthermore, only one out of five dogs with pre-capillary PH showed improvement in RHF. These relatively poor responses in right heart morphology and clinical symptoms associated with PH suggest that 15 µg/kg BPS might be insufficient for treating dogs with progressed pre-capillary PH. Since a dose-dependent pulmonary vasodilatory effect was observed up to 25 µg/kg BPS in canine models of chronic embolic PH [15], a higher dose of BPS might be more effective. Furthermore, a combination with other pulmonary vasodilators (e.g., sildenafil) might be expected to have an additive effect. Further studies investigating a higher dose of BPS and its combination with other pulmonary vasodilators are expected to evaluate the optimal treatment for dogs with progressive pre-capillary PH.

This study had several limitations. First, we could not perform right heart catheterization for a definitive diagnosis of PH. Furthermore, post-capillary PH was classified based on echocardiographic evidence of an increased left heart size and increased left atrial pressure. Dogs with post-capillary PH in this study might have had a pre-capillary PH factor. Second, there was a bias in the causative factors of PH; PH in most dogs was caused by myxomatous mitral valve disease (left heart disease) or respiratory diseases. Further studies are needed to evaluate the efficacy of BPS in canine PH with causes other than left heart and respiratory diseases. Third, we were unable to standardize medications. However, because medications other than BPS were not modified during the study period, we believe that this study could evaluate the effect of BPS with minimal influence from other drugs. Fourth, the small sample size of this study affected the statistical analyses performed to detect differences between groups. Fifth, in this study, the frequency of BPS administration was twice daily based on previous studies [15,19]. However, in humans, BPS is commonly administered three times per day owing to its relatively short half-life. Therefore, frequent administration of BPS may be more effective in dogs with PH. Finally, we could not compare the efficacy of BPS and sildenafil. Further study comparing the clinical efficacy of BPS and sildenafil would be expected in the future.

## 5. Conclusions

In this study, improvements in clinical signs related to PH were observed in some cases. Additionally, BPS decreased PVI and TR velocities through pulmonary vasodilation in dogs with PH, regardless of the causative disease. Consequently, the RV function and pulmonary circulation improved significantly in dogs with PH. These results suggest that BPS could be an additional treatment option for dogs with PH induced by various causes. Especially in dogs with post-capillary PH, which is the most common cause of PH in veterinary medicine, BPS could be an optimal option to treat PH because of its low risk of worsening left atrial pressure. These preliminary results should be confirmed by multicentric, prospective, randomized, placebo-controlled, blinded studies with clinical endpoints (e.g., the survival time) to understand the real clinical efficacy of BPS alone or in combination with sildenafil in dogs with PH.

## Figures and Tables

**Figure 1 animals-12-02078-f001:**
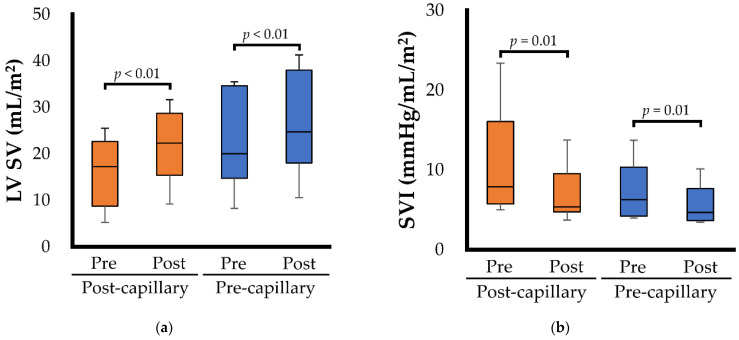
Box and whisker plots showing the specific right heart hemodynamics and functional echocardiographic variables before (Pre) and after (Post) beraprost sodium administration. The bottom of the box is 25th percentile, the center line is the median, the top of the box is 75th percentile, and the whiskers are the range. The blue box represents the results for dogs with pre-capillary pulmonary hypertension; the orange box represents the results for dogs with post-capillary pulmonary hypertension. (**a**) Left ventricular stroke volume (LV SV); (**b**) Systemic vascular impedance (SVI); (**c**) Left ventricular longitudinal strain (LV-SL); (**d**) Left ventricular circumferential strain (LV-SC).

**Figure 2 animals-12-02078-f002:**
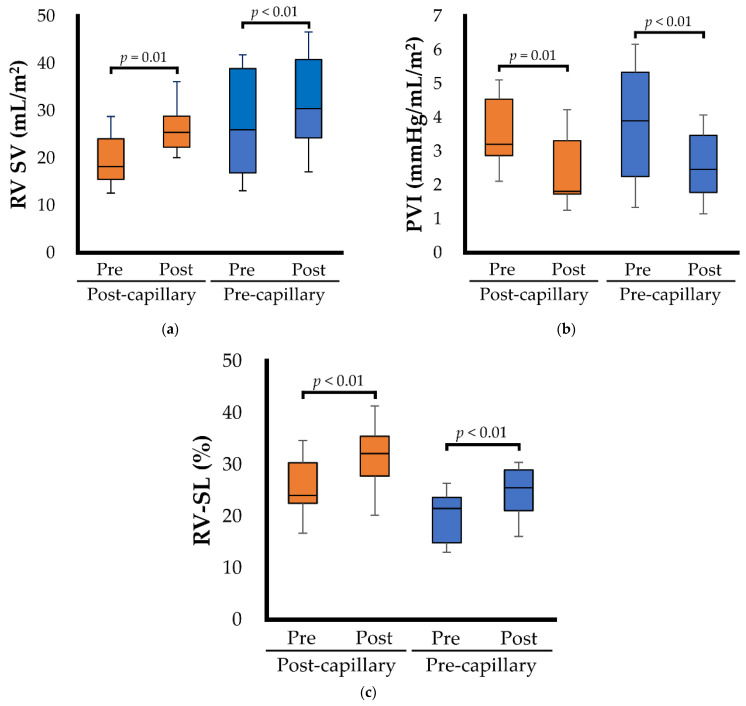
Box and whisker plots showing the specific right heart hemodynamics and functional echocardiographic variables before (Pre) and after (Post) beraprost sodium administration. The bottom of the box is 25th percentile, the center line is the median, the top of the box is 75th percentile, and the whiskers are the range. The blue box represents the results for dogs with pre-capillary pulmonary hypertension; the orange box represents the results for dogs with post-capillary pulmonary hypertension. (**a**) Right ventricular stroke volume (RV SV); (**b**) Pulmonary vascular impedance (PVI); (**c**) Right ventricular longitudinal strain (RV-SL).

**Table 1 animals-12-02078-t001:** Clinical characteristics of dogs with pulmonary hypertension before and after beraprost sodium administration.

Variables	Pre-Examination	Post-Examination
Age (year)	13.2 (10.9–14.6)	13.3 (10.9–14.7)
Sex (male/female)	3/13	
Body weight (kg)	4.0 (3.2–6.7)	4.0 (3.1–6.7)
BPS dose (µg/kg)	Range: 13.4–22.0	
Dosing period (days)	Range: 7–21	
Heart rate (bpm)	125 (109–148)	121 (107–138)
Systolic blood pressure (mmHg)	136 (123–144)	125 (114–141)
Mean blood pressure (mmHg)	100 (86–111)	98 (82–107)
PH severity (mild/moderate/severe)	0/5/11	0/7/9
RHF (n)		
Pre-capillary PH	3/8 (38%)	2/8 (25%)
Post-capillary PH	5/8 (63%)	2/8 (25%)
Clinical signs suggestive of PH (n)		
Syncope	10/16 (63%)	7/16 (44%)
Weakness	13/16 (81%)	9/16 (56%)
Respiratory distress at rest	6/16 (38%)	6/16 (38%)
Exercise intolerance	11/16 (69%)	8/16 (50%)

Abbreviations: BPS, beraprost sodium; PH, pulmonary hypertension; RHF, right heart failure.

**Table 2 animals-12-02078-t002:** Medications which 16 dogs with PH received at the time of examination.

Medications	Post-Capillary PH	Pre-Capillary PH
Angiotensin-converting enzyme inhibitor (n)	8/8 (100%)	1/8 (13%)
Pimobendan (n)	8/8 (100%)	2/8 (25%)
Spironolactone (n)	4/8 (50%)	0/8 (0%)
Loop diuretics (n)	6/8 (75%)	0/8 (0%)
Calcium channel blocker (n)	4/8 (50%)	0/8 (0%)
Isosorbide dinitrate (n)	6/8 (75%)	0/8 (0%)
Sildenafil (n)	3/8 (38%)	2/8 (25%)
Antimicrobials (n)	0/8 (0%)	5/8 (63%)
Corticosteroid (n)	0/8 (0%)	5/8 (63%)
Antiplatelet drug (n)	0/8 (0%)	1/8 (13%)

**Table 3 animals-12-02078-t003:** Results of echocardiographic variables for left heart morphology and function in dogs with post-capillary and pre-capillary pulmonary hypertension before and after beraprost sodium administration.

Variables	Post-Capillary PH (*n* = 8)			Pre-Capillary PH (*n* = 8)		
	Pre-Examination	Post-Examination	*p*	Pre-Examination	Post-Examination	*p*
LA/Ao	2.2 (1.7–2.9)	2.4 (1.6–2.7)	0.093	1.0 (1.0–1.2)	1.1 (1.0–1.3)	0.310
LVIDDN (cm/kg^0.294^)	2.1 (1.6–2.2)	2.3 (1.6–2.4)	0.211	1.1 (0.8–1.3)	1.2 (0.9–1.4)	0.327
LVIDSN (cm/kg^0.315^)	0.9 (0.9–1.1)	1.0 (0.8–1.2)	0.465	0.6 (0.5–0.6)	0.6 (0.5–0.8)	0.533
Fractional shortening (%)	48.3 (41.5–54.4)	49.8 (41.5–57.4)	0.579	45.3 (38.3–52.5)	46.2 (42.9–53.9)	0.150
LVEDVI (mL/m^2^)	79.0 (53.6–97.7)	88.0 (54.5–105.2)	0.130	28.2 (16.1–37.9)	32.3 (19.7–39.0) *	0.032
LVESVI (mL/m^2^)	27.9 (23.1–40.9)	27.9 (20.5–38.1)	0.251	13.0 (8.1–18.4)	12.8 (10.5–14.5)	0.515
Ejection fraction (%)	61.1 (51.0–65.3)	64.6 (58.1–70.6) *	0.004	51.6 (46.2–54.5)	61.2 (48.2–64.0) *	0.024
E (m/s)	1.2 (1.0–1.5)	1.2 (0.8–1.3)	0.354	0.6 (0.5–0.6)	0.5 (0.5–0.6)	0.594
E/A	1.3 (1.0–2.5)	1.2 (0.8–1.5)	0.362	0.8 (0.7–1.0)	0.8 (0.8–0.9)	0.407
E/e’	18.5 (12.9–23.3)	12.1 (10.1–18.0)	0.230	13.4 (8.9–15.3)	13.6 (11.1–14.7)	0.906
MR velocity (m/s)	5.5 (5.2–5.8)	5.9 (5.5–6.2) *	0.047	Not available	Not available	

Data are expressed as the median as well as the 25th–75th percentile values. * The value was significantly different compared to the pre-examination value (*p* < 0.05). Abbreviations: A, late diastolic transmitral flow velocity; E, early diastolic transmitral flow velocity; e’, peak early diastolic myocardial velocity at the septal mitral annulus; LA/Ao, left atrial to aortic diameter ratio; LVEDVI, end-diastolic left ventricular volume normalized by body surface area; LVESVI, end-systolic left ventricular volume normalized by body surface area; LVIDDN, end-diastolic left ventricular internal dimension normalized by body weight; LVIDSN, end-systolic left ventricular internal dimension normalized by body weight; MR, mitral regurgitation.

**Table 4 animals-12-02078-t004:** Results of echocardiographic variables for right heart morphology and function in dogs with post-capillary and pre-capillary pulmonary hypertension before and after beraprost sodium administration.

Variables	Post-Capillary PH (*n* = 8)			Pre-Capillary PH (*n* = 8)		
	Pre-Examination	Post-Examination	*p*	Pre-Examination	Post-Examination	*p*
PA/Ao	1.0 (1.0–1.1)	1.0 (0.9–1.1)	0.635	1.0 (1.0–1.1)	1.0 (1.0–1.1)	0.331
RVEDA (cm^2^)	2.9 (1.8–4.7)	2.8 (1.8–3.6)	0.371	3.4 (2.0–4.2)	3.3 (2.3–4.0)	0.221
RVESA (cm^2^)	1.4 (1.0–2.9)	1.4 (0.9–1.7)	0.111	2.2 (1.1–2.6)	1.7 (1.1–2.3)	0.050
RV FACn (%/kg^−0.097^)	46.4 (35.6–54.7)	59.5 (55.9–63.0) *	0.006	45.9 (39.4–50.5)	48.5 (43.0–57.8) *	0.034
TAPSEn (mm/kg^0.284^)	6.6 (6.3–8.1)	7.4 (6.2–9.3)	0.374	4.1 (3.0–5.2)	5.0 (4.5–5.5)	0.150
RV s’ (cm/s)	13.2 (8.4–16.2)	11.6 (9.4–17.9)	0.627	5.3 (4.9–7.6)	7.3 (5.4–8.9)	0.164
TR velocity (m/s)	4.1 (3.6–5.0)	3.5 (3.1–4.9) *	0.033	4.6 (4.5–5.2)	4.2 (4.1–4.8) *	0.004

Data are expressed as the median as well as the 25th–75th percentile values. * The value was significantly different compared to the pre-examination value (*p* < 0.05). Abbreviations: PA/Ao, pulmonary artery to aortic diameter ratio; RV FACn, right ventricular fractional area change normalized by body weight; RV s’, peak systolic myocardial velocity of the lateral tricuspid annulus; RVEDA, end-diastolic right ventricular area; RVESA, end-systolic right ventricular area; TAPSEn, tricuspid annular plane systolic excursion normalized by body weight; TR, tricuspid regurgitation.

## Data Availability

The datasets used or analyzed during the current study are available from the corresponding author upon reasonable request.

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
