# Peer review of "Beraprost Sodium for Pulmonary Hypertension in Dogs: Effect on Hemodynamics and Cardiac Function"

_animals, 2022, doi:10.3390/ani12162078_

Round 1
Reviewer 1 Report
Thus, this study aimed to investigate the clinical efficacy of BPS in dogs with PH of 60 various etiologies.
Author Response
Dear Reviewer 1
We wish to express our strong appreciation to the Reviewer for reviewing our manuscripts.
Reviewer 2 Report
The manuscript entitled “Beraprost sodium for pulmonary hypertension in dogs: effect on hemodynamics and cardiac function” describes the clinical and cardiovascular response to the use of Beraprost sodium in dogs with pulmonary hypertension, opening new perspectives for the therapy of PH in dogs. It is a very good study that can be published in the Animals journal.
I suggest the following minor revisions to the authors:
SIMPLE SUMMARY
Line 13: I suggest remove the word "or" from the sentence "pulmonary arterial pressure and/or pulmonary vascular resistance". In fact, resistance and pressure are two directly proportional and strictly interdependent variables. Alternatively, the sentence "and / or pulmonary vascular resistance" could be removed and left only the definition of pulmonary arterial hypertension.
Line 16: I don’t know the reality of the geographical area in which the authors work but in Europe Sildenafil is easily available on the market. Please specify what is meant by the sentence "difficulty in obtaining the drug"
Line 16: it is more correct to write "inter-individual variability in the response to sildenafil therapy" rather than "drug resistance".
Lines 18-20: it is more correct to write “In this study, beraprost sodium showed significant pulmonary and systemic vasodilation without any adverse effects in sixteen dogs with pulmonary hypertension”
Lines 21-22: it is more correct to write “These results emphasize the potential efficacy of beraprost sodium in treating canine pulmonary hypertension”.
ABSTRACT
Line 25: it is more correct to write “the clinical and cardiovascular response” rather than “the clinical efficacy”
INTRODUCTION
Lines 49-50: what do you mean with the word inaccessibility? Please specify. The sentence “potential drug resistance” should be replaced by “inter-individual variability in the therapy response”
Lines 50-51: the sentence “Furthermore, in dogs with post-capillary PH, increased pulmonary circulation may elevate the already elevated left atrial pressure [8–10].” It’s correct but the references that the authors cite refers to human medicine and not to canine medicine, please check.
Line 60: it is more correct to write “the clinical and cardiovascular response” rather than “the clinical efficacy”
MATERIALS AND METHODS
Line 77: please specify which device did you use for oscillometric systemic pressure measurements.
Lines 121-139: please provide adequate references of each single measurement and formula used in the study in order to allow readers to explore these topics.
Line 122: I think there is a typo: e1 is the velocity of the septal mitral annulus at early-diastole. Please verify.
RESULTS
Lines 164-165 and table 1: in this section the authors should specify in detail how many dogs had ascites, pleural effusion, pericardial effusion, or syncope, without any abnormalities other than PH, before and after BPS therapy. It would also be useful to specify if dogs with post-capillary PH were being treated with any drug and if so with which one.
Lines 168-169: in which ACVIM class (B-C or D) were the dogs with post-capillary PH? Were they under concomitant treatment during the study period? If yes, which concomitant treatment?
DISCUSSION
Lines 255-256: the sentence “In dogs with post-capillary PH, improved pulmonary circulation may burden the left heart and increase elevated left atrial pressure [8].” It’s correct but the references that the authors cite refers to human medicine and not to canine medicine, please check.
Lines 261-262 how do you explain that a reduction in SVI and systemic pressure, due to the vasodilatory effect of BPS, resulted in a significant increase in MR velocity in dogs with post-capillary PH? One cause could be the increase in systolic contraction of the LV highlighted by speckle tracking results, this should be discussed.
In the conclusions and /or discussions the authors should say that these preliminary results should be confirmed by multicentric, prospective, randomized, placebo-controlled, blinded studies with clinical endpoints (eg. the survival time) to understand the real clinical efficacy of BPS alone or in combination with sildenafil in dogs with PH.
Author Response
Dear Reviewer 2
Comments and Suggestions for Authors
The manuscript entitled “Beraprost sodium for pulmonary hypertension in dogs: effect on hemodynamics and cardiac function” describes the clinical and cardiovascular response to the use of Beraprost sodium in dogs with pulmonary hypertension, opening new perspectives for the therapy of PH in dogs. It is a very good study that can be published in the Animals journal.
I suggest the following minor revisions to the authors:
Response: We wish to express our strong appreciation to the Reviewer for their insightful comments on our paper. We feel the comments have helped us significantly improve the paper. We hope that the revised paper meets your approval and will be more suitable for publication in the Animals journal for the Article. Please see the following point-by-point responses for details.
SIMPLE SUMMARY
Line 13: I suggest remove the word "or" from the sentence "pulmonary arterial pressure and/or pulmonary vascular resistance". In fact, resistance and pressure are two directly proportional and strictly interdependent variables. Alternatively, the sentence "and / or pulmonary vascular resistance" could be removed and left only the definition of pulmonary arterial hypertension.
Response: Thank you very much for your suggestion. We have changed the word “and/or” into the “and” according to the Reviewer’s comment.
Line 12-13: Pulmonary hypertension is potentially life-threatening disease among dogs that is characterized by increased pulmonary arterial pressure and pulmonary vascular resistance.
Line 16: I don’t know the reality of the geographical area in which the authors work but in Europe Sildenafil is easily available on the market. Please specify what is meant by the sentence "difficulty in obtaining the drug"
Response: Thank you very much for your comment. In Japan, obtaining genuine medicine of the sildenafil requires importation from abroad, which might take a lot of trouble. As you mentioned, the difficulty in obtaining the drug may be limited to some areas. We have modified the sentence as follows.
Line 15-17: However, the availability of sildenafil is limited because of its high cost, difficulty in obtaining the drug in some area, and potential inter-individual variability in the response to sildenafil therapy.
Line 16: it is more correct to write "inter-individual variability in the response to sildenafil therapy" rather than "drug resistance".
Response: Thank you very much for your advice. We have modified the words according to the Reviewer’s comment.
Line 15-17: However, the availability of sildenafil is limited because of its high cost, difficulty in obtaining the drug in some area, and potential inter-individual variability in the response to sildenafil therapy.
Lines 18-20: it is more correct to write “In this study, beraprost sodium showed significant pulmonary and systemic vasodilation without any adverse effects in sixteen dogs with pulmonary hypertension”
Response: Thank you very much for your comment. We have corrected the sentence according to the Reviewer’s comment.
Line 19-20: In this study, beraprost sodium showed significant pulmonary and systemic vasodilation without any adverse effects in sixteen dogs with pulmonary hypertension.
Lines 21-22: it is more correct to write “These results emphasize the potential efficacy of beraprost sodium in treating canine pulmonary hypertension”.
Response: Thank you very much for your advice. We have corrected the sentence according to the Reviewer’s comment.
Line 22-23: These results emphasize the potential efficacy of beraprost sodium in treating canine pulmonary hypertension.
ABSTRACT
Line 25: it is more correct to write “the clinical and cardiovascular response” rather than “the clinical efficacy”
Response: Thank you very much for your comment. We have corrected the sentence according to the Reviewer’s comment.
Line 25-27: This study aimed to evaluate the clinical and cardiovascular response of BPS in canine patients with PH of various causes.
INTRODUCTION
Lines 49-50: what do you mean with the word inaccessibility? Please specify. The sentence “potential drug resistance” should be replaced by “inter-individual variability in the therapy response”
Response: Thank you very much for your comment. We wrote the word “inaccessibility” to imply the difficulty in obtaining the drug in some area. We have modified the sentence according to the Reviewer’s comment.
Line 50-52: However, the availability of sildenafil is limited because of its high cost, difficulty in obtaining the drug in some area, and potential inter-individual variability in the response to sildenafil therapy due to genetic polymorphism [7].
Lines 50-51: the sentence “Furthermore, in dogs with post-capillary PH, increased pulmonary circulation may elevate the already elevated left atrial pressure [8–10].” It’s correct but the references that the authors cite refers to human medicine and not to canine medicine, please check.
Response: Thank you very much for your comment. As you mentioned, these references refer to human medicine. We have corrected the sentences as follows according to the Reviewer 2 and 3’s comments.
Line 52-55: Furthermore, in dogs and human patients with post-capillary PH, sildenafil-induced in-creased pulmonary circulation may elevate the already high left atrial pressure with the consequent risk of precipitating left congestive heart failure or worsening an ongoing lung edema [1,8–10].
Line 60: it is more correct to write “the clinical and cardiovascular response” rather than “the clinical efficacy”
Response: Thank you very much for your comment. We have corrected the sentence according to the Reviewer’s comment.
Line: Thus, this study aimed to investigate the clinical and cardiovascular response of BPS in dogs with PH of various etiologies.
MATERIALS AND METHODS
Line 77: please specify which device did you use for oscillometric systemic pressure measurements.
Response: Thank you very much for your comment. We have added the information about the device used for oscillometric systemic pressure measurements.
Line 81-84: All dogs underwent complete physical examination, oscillometric method-derived blood pressure measurement (BP100DII, FUKUDA M-E KOGYO Co, LTD, Tokyo, Japan), 6-leads electrocardiography, and radiographic and echocardiographic examinations. Electrocardiography was performed for 30 seconds at least twice.
Lines 121-139: please provide adequate references of each single measurement and formula used in the study in order to allow readers to explore these topics.
Response: Thank you very much for your advice. As you mentioned, some echocardiographic variables not include references. We have added the references into each echocardiographic variable.
Line 136-138: Tissue Doppler imaging-derived peak myocardial velocity of the septal mitral annulus at early-diastole (e’) was also measured [26]. Using these Doppler echocardiographic variables, E/A and E/e’ were calculated [27].
Line 122: I think there is a typo: e1 is the velocity of the septal mitral annulus at early-diastole. Please verify.
Response: Thank you very much for your comment, and we apologize the erroneous description. We have corrected the word as follows.
Line 136-137: Tissue Doppler imaging-derived peak myocardial velocity of the septal mitral annulus at early-diastole (e’) was also measured [26].
RESULTS
Lines 164-165 and table 1: in this section the authors should specify in detail how many dogs had ascites, pleural effusion, pericardial effusion, or syncope, without any abnormalities other than PH, before and after BPS therapy. It would also be useful to specify if dogs with post-capillary PH were being treated with any drug and if so with which one.
Response: Thank you very much for your advice. We have added the detailed information about the RHF in the results. Additionally, we have added the table representing the medications which 16 dogs with PH received according to the Reviewer 2 and 3’s comment.
Line 192-195: The clinical findings indicating the RHF observed before the BPS administration in dogs with pre- and post-capillary PH were follows: pre-capillary PH, ascites (n = 2), ascites and pleural effusion (n = 1); post-capillary PH, ascites (n = 2), ascites and pericardial effusion (n = 2), pericardial effusion (n= 1).
Lines 168-169: in which ACVIM class (B-C or D) were the dogs with post-capillary PH? Were they under concomitant treatment during the study period? If yes, which concomitant treatment?
Response: Thank you very much for your comment. Almost all dogs with post-capillary PH in this study showed ACVIM stage C or D, and only one dog with post-capillary PH showed ACVIM stage B2. Additionally, we have added the table representing the medications which 16 dogs with PH received according to the Reviewer 2 and 3’s comment.
Line 183-185: All cases of post-capillary PH were caused by myxomatous mitral valve disease (ACVIM stage: one dog was identified as stage B2 and the others were stage C or D).
DISCUSSION
Lines 255-256: the sentence “In dogs with post-capillary PH, improved pulmonary circulation may burden the left heart and increase elevated left atrial pressure [8].” It’s correct but the references that the authors cite refers to human medicine and not to canine medicine, please check.
Response: Thank you very much for your comment. As you mentioned, these references refer to human medicine. We have corrected the sentences as follows.
Line 286-287: In human patients with post-capillary PH, improved pulmonary circulation may burden the left heart and increase elevated left atrial pressure [8].
Lines 261-262 how do you explain that a reduction in SVI and systemic pressure, due to the vasodilatory effect of BPS, resulted in a significant increase in MR velocity in dogs with post-capillary PH? One cause could be the increase in systolic contraction of the LV highlighted by speckle tracking results, this should be discussed.
Response: Thank you very much for your advice. As you mentioned, LV systolic function based on LV-SL might also influence the increase in MR velocity. We have added the following sentence into the discussion.
Line 300-301: Additionally, improved LV systolic function might also influence the increase in MR velocity.
In the conclusions and/or discussions the authors should say that these preliminary results should be confirmed by multicentric, prospective, randomized, placebo-controlled, blinded studies with clinical endpoints (eg. the survival time) to understand the real clinical efficacy of BPS alone or in combination with sildenafil in dogs with PH.
Response: Thank you very much for your advice. We have added the sentences into the conclusions according to the Reviewer’s comment.
Line 347-350: These preliminary results should be confirmed by multicentric, prospective, randomized, placebo-controlled, blinded studies with clinical endpoints (e.g., the survival time) to understand the real clinical efficacy of BPS alone or in combination with sildenafil in dogs with PH.

Reviewer 3 Report
Comments to the Authors are in the attached file

Author Response
Dear Reviewer 3
Comments and Suggestions for Authors
The manuscript descripts the use of a novel therapy in dogs with naturally acquired pulmonary hypertension. The article is interesting and well organized, and the topic innovative. Moreover, the findings form this report may pave the wave for future researches on this field/drug that and may be clinically relevant is expanded further. However, I have some comments and suggestion aimed at correct some imprecisions and improved the clinical soundness of the results of this research.
Response: We wish to express our strong appreciation to the Reviewer for their insightful comments on our paper. We feel the comments have helped us significantly improve the paper. We hope that the revised paper meets your approval and will be more suitable for publication in the Animals journal for the Article. Please see the following point-by-point responses for details.
INTRODUCTION
-lines 49-50: “However, the availability of sildenafil is limited because of its high cost, inaccessibility, and potential drug resistance”. I suggest to write between parenthesis the daily cost for a 10 kg dog at a standard, like dose of 2 mg/kg PO BID, for sildenafil, so that readers can understand what mean “high cost”. This will be particularly important also because, later, I will ask you to report the cost for Beraprost, so that readers can compare the cost of two protocols. Moreover, please explain what means potential drug resistance (I think you are referring to the one related to genetic polymorphism, but this should be made explicit for readers).
Response: Thank you very much for your comments. In our institution, the delivered price is 1,242.5 Japanese yen per tablet for sildenafil and 66 Japanese yen per tablet for beraprost sodium. Therefore, in our institution, the price of 2.0 mg/kg sildenafil for a 10 kg dog would be 2.8 times higher than that of 15 µg/kg of beraprost sodium for a 10 kg dog. However, as for the selling price, we consider it difficult to state the price because of the possibility of errors from one facility to another. Additionally, we have explained the potential drug resistance in detail (the words “drug resistance” were modified according to the Reviewer 2’s comment).
Line 50-52: However, the availability of sildenafil is limited because of its high cost, difficulty in obtaining the drug in some area, and potential inter-individual variability in the response to sildenafil therapy due to genetic polymorphism [7].
-lines 50-51: “. Furthermore, in dogs with post-capillary PH, increased pulmonary circulation may elevate the already elevated left atrial pressure” I suggest to add this part a part to this sentence so that the final result would be: “Furthermore, in dogs with post-capillary PH, sildenafil-induced increased pulmonary circulation may elevate the already high left atrial pressure with the consequent risk of precipitating left congestive heart failure or worsening an ongoing lung edema.” Moreover, in addition to the references you have put at the end of this sentence, I would also put the reference related to the PH guidelines.
Response: Thank you very much for your advice. We have changed the sentence according to the Reviewer’s comment. Additionally, because the references which we put the end of the sentence refers human medicine, we have modified the sentences as follows.
Line 52-55: Furthermore, in dogs and human patients with post-capillary PH, sildenafil-induced increased pulmonary circulation may elevate the already high left atrial pressure with the consequent risk of precipitating left congestive heart failure or worsening an ongoing lung edema [1,8–10]. 
MATERIALS AND METHODS
-line 78: A question related to electrocardiography: which technique did you used? 1-lead? 6-lead? 12-lead? For how many minutes? 1, 2, 3 or more? Did you also perform an Holter in some dogs? Please provide addition information at regard.
Response: Thank you very much for your comment. In this study, dogs were performed 30-second 6-lead electrocardiography more than twice. We have added the information above into the materials and methods.
Line 81-84: All dogs underwent complete physical examination, oscillometric method-derived blood pressure measurement (BP100DII, FUKUDA M-E KOGYO Co, LTD, Tokyo, Japan), 6-leads electrocardiography, and radiographic and echocardiographic examinations. Electrocardiography was performed for 30 seconds at least twice.
-lines 79-83: “We clinically diagnosed PH based on a high probability of PH noted in the American College of Veterinary Internal Medicine (ACVIM) consensus statement, which was defined using echocardiographic findings of tricuspid regurgitation (TR) velocity and anatomical abnormalities of the right heart, pulmonary artery, and caudal vena cava”. Please, introduce a cut-off related to the peak velocity of TR that you used to define PH. Moreover, I would introduce also the cut-offs by which you define PH as mild, moderate and severe.
Response: Thank you very much for your suggestions. This study used the peak TR velocity > 3.4 m/s to diagnose PH. Additionally, dogs with PH were classified into one of three PH severity groups (mild, moderate, severe) according to the TR velocity and the presence of right heart failure.
Line 84-88: We clinically diagnosed PH based on a high probability of PH noted in the American College of Veterinary Internal Medicine (ACVIM) consensus statement, which was defined using echocardiographic findings of tricuspid regurgitation (TR) velocity (> 3.4 m/s) and anatomical abnormalities of the right heart, pulmonary artery, and caudal vena cava [1].
Line 95-100: In this study, the severity of PH was estimated according to the TR velocity [16–18]. Mild PH was identified if dogs had TR velocity < 3.5 m/s without any evidence of RHF; moderate PH was identified if dogs TR velocity 3.5–4.3 m/s without any evidence of RHF; severe PH was identified if dogs had TR velocity > 4.3 m/s [16–18]. Additionally, dogs with RHF were considered as having severe PH regardless of the TR velocity.
-lines 85-88: “Furthermore, dogs were determined to have RHF if they exhibited at least one clinical evidence of ascites, pleural effusion, and/or mild pericardial effusion, or a clinical sign of syncope without any abnormalities other than PH that might have been responsible [15,16]” I do not agree with the use of the term heart failure in these dogs because for us, in veterinary medicine, is more likely to be diagnosed a ‘congestive’ heart failure. Therefore, I suggest to change the definition here and later in the text. I also suggest to delete the word ‘mild’ before pericardial effusion, as the effusion in dogs with RCHF may be also moderate or severe. Lastly, I suggest to not include syncope in the list of signs related to RCHF as it is mainly related to the hypoperfusion of LV due to severe PH, with consequent decreased CO, rather an enlarged and congested RV due to PH.
Response: Thank you very much for your comment. As you mentioned, cardiac syncope is not the clinical sign of “congestive” heart failure. We have excluded the syncope from the definition of RHF. Additionally, we have added the results of clinical signs suggestive of PH including syncope. Please see the main document for the details.
Line 91-95: Furthermore, dogs were determined to have RHF if they exhibited at least one clinical evidence of ascites, pleural effusion, and/or pericardial effusion without any abnormalities other than PH that might have been responsible [15,16]. We also questioned the dogs’ owner about the clinical signs suggestive of PH, such as syncope, respiratory distress at rest, and exercise intolerance [1].
-lines 91-92: “All dogs were orally administered BPS twice daily using 55-μg tablets (Toray Industries, Inc., Tokyo, Japan), divided into doses of approximately 15 μg/kg [15,17]” I would specify that the dosage is derived form an experimental study on a canine model of PH and a clinical study on cats with chronic kidney disease, so that readers may known the origin of such a dose.
Response: Thank you very much for your advice. We have added the information about how to determine the dosage and frequency of BPS in this study according to the Reviewer’s comment.
Line 104-106: The dosage and frequency of BPS were determined according to an experimental study on a canine model of PH and a clinical study on cats with chronic kidney disease [15,17].
-line 108: what do you mean for “cardiologist”. Is this operator a board-certified cardiologist by ECVIM-CA or ACVIM? If not, I would delete that word and I would use something similar to “an operator with years of experience in echocardiography”.
Response: Thank you very much for your comment, Because the operator is not a board-certified cardiologist by ECVIM-CA or ACVIM, we have modified the word according to the Reviewer’s comment.
Line 120-123: All echocardiographic measurements were performed by another observer (Y.Yu.) who had been trained by an operator with years of experience in echocardiography using an offline workstation (EchoPAC PC, version 204; GE Healthcare, Tokyo, Japan).
RESULTS
-In the section of clinical characteristics of the study population, in addition to the types of PH and the causes, I suggest to add also the degree of severity of PH. Indeed, a question can be: were all these dogs affected by severe PH? A second question, that merits further explanation son this section, is: how many dogs were symptomatic for PH? Indeed, you say that 10 dogs out 16 had RHF? Does this mean that only 10/16 dogs were symptomatic for PH?
This is important because only dogs with severe PH or dogs with symptomatic PH really need a symptomatic treatment with a vasodilator. I also say this because if I read Table 3, then I see that values of TR ranged from 3.6 to 5 m/s, and I strongly believe that give a pulmonary vasodilator in a dog with 3.6 m/s, especially if symptomatic and/or due to advance mitral valve disease may be, in many similar cases, a clinical mistake.
Please, provide more data on these aspects, including: 1) the degree of PH, 2) the number of dogs for each category of severity of PH, 3) the number of symptomatic dogs, 4) the type of clinical signs. Then, also Table 1 should be expanded with similar data.
Moreover, another lacking information is the one related to other treatment. Please, create a new table with the name of each drug, the number of dogs receiving it and the dosage. Moreover, it is important to specify in some drugs have been added or deleted during the study protocol. If yes, you should specify which and in how many dogs. Lastly, it should be specified if the dose of some drug has been changed during the study protocol. In this case, you should specify the range of change the name of the drug and the number of dogs in which this change occurred.
Response: Thank you very much for your advices. According to the Reviewer’s comment, we have added the results about the severity of PH. In this study, we have classified dogs with PH into one of the three PH severity groups based on the tricuspid regurgitation velocity and the presence of right heart failure. Additionally, according to the Reviewer 2 and 3’s comment, we have added the table representing the medications which dogs with PH used in this study received. Described in the study protocol (Line: 108-109), there were no changes in medications other than BPS during the study period in all dogs with PH. To help readers better understand, we have also added the contents into the results.
Line 186-187: There were no changes in medications other than BPS during the study period in all dogs with PH.
Line 192-196: The clinical findings indicating the RHF observed before the BPS administration in dogs with pre- and post-capillary PH were follows: pre-capillary PH, ascites (n = 2), ascites and pleural effusion (n = 1); post-capillary PH, ascites (n = 2), ascites and pericardial effusion (n = 2), pericardial effusion (n= 1). All dogs had at least one clinical sign suggestive of PH at the pre-examination.
-lines 165-166: “The causes of pre-capillary PH were hypoxia and other respiratory diseases (n = 5),” I suggest to rewrite this sentence in this way: “The causes of pre-capillary PH were hypoxia due to various respiratory diseases (n = 5),”
Response: Thank you very much for your suggestion. We have corrected the sentence according to the Reviewer’s suggestion.
Line 180-183: The causes of pre-capillary PH were hypoxia due to various respiratory diseases (n = 5), reversed patent duct arteriosus (n = 1), filariasis (n = 1), and chronic pulmonary thromboembolic disease attributed to hyperadrenocorticism (n = 1).
Table 2: I suggest to add a column on the right of “post examination” for both pre- and post-capillary PH and show the relative P-values of each parameter to readers more clearly. Then, if you show all P-values in Table, you can delete them form the text so that the same information is not repeated twice.
Response: Thank you very much for your suggestion. We have added the P values of each echocardiographic variables in Table 2 and 3 according to the Reviewer’s suggestion. Please see the Table 2 and 3 for the details.
-A lacking part of results is the one related to clinical signs: in how many dogs the clinical signs improved and disappeared? Indeed, for a drug used to treat PH, it is more important to alleviate clinical signs than increase/decrease some echocardiographic parameters, as we are treating patients and not numbers. Indeed, I can also see a statistically significant increase/decrease of an echocardiographic measurement, but if my patient is not clinically improved that drugs remains, anyway, not clinically efficient.
In the light of my comments, the Results section overall needs to be expanded with more data on the clinical conditions (before and post-treatment) and therapies of your patients. Without such data the results remain not really relevant for the scientific community.
Response: Thank you very much for your comment. As you mentioned, there was no results about the clinical signs other than RHF. In this study, all dogs showed at least one clinical sign suggestive of PH at the time of pre-examination, and clinical improvements were observed in some cases. We have added the results related to the clinical signs into the results and table. Please see the main document for the details.
DISCUSSION
-lines 240-242: These results suggest that BPS may be an additional treatment option for canine patients with PH, regardless of the causative diseases” I suggest to remove the word “additional” as it may be misunderstood by readers (they may think that it has been that is a good/safe option to add Beraprost to convention therapies, like sildenafil, although this has not been investigated). Moreover, the fact that this drug is really useful in dogs with PH mainly depends on its clinical effects rather than the echocardiographic ones. Therefore, in my opinion, you can maintain such a strong affirmation only after having expanded the results section with clinical data and having demonstrated a significant beneficial clinical effect of the drug.
Response: Thank you very much for your suggestion. As you mentioned, we did not investigate whether adding BPS to conventional therapies such as sildenafil is a good/safe option. We have deleted the word “additional” according to the Reviewer’s comment. Additionally, we have added the results of clinical signs related to PH. We consider that BPS would have beneficial effect on canine PH based on the clinical and echocardiographic improvements observed in this study. Please see the main document for the details.
Line 269-271: These results suggest that BPS may be a treatment option for canine patients with PH, regardless of the causative diseases.
-line 247: “BPS decreased PVI and RV pressure overload based on the TR velocity” this sounds like a repletion as you have already said this at the beginning of discussion.
Response: Thank you very much for your comment. As you mentioned, this sentence sounds like a repletion. We have reworded the sentence as follows.
Line 275-276: BPS significantly improved RV hemodynamics and loading conditions based on the decrease in PVI and TR velocity.
-lines 252-254: please rewrite this sentence in this way “Additionally, although we did not evaluate the antiplatelet effect of BPS, it may also be effective, especially in dogs with PH with a potential for causing pulmonary thromboembolism. However, it is important to consider that further investigations are needed to properly characterize the antiplatelet action of this molecule in naturally-acquire thromboembolic diseases in dogs.”
Response: Thank you very much for your suggestion. We have modified the sentence according to the Reviewer’s suggestion.
Line 281-285: Additionally, although we did not evaluate the antiplatelet effect of BPS, it may also be effective, especially in dogs with PH with a potential for causing pulmonary thromboembolism. However, it is important to consider that further investigations are needed to properly characterize the antiplatelet action of this molecule in naturally-acquire thromboembolic diseases in dogs.
-lines 273-275: “The improved LV function in dogs with pre-capillary PH was possibly due to increased LV filling (i.e., increased LVEDVI), in addition to the systemic vasodilating effect of BPS” I would rewrite this sentence by adding a part, so that readers could really understand your theory: “The improved LV function in dogs with pre-capillary PH was possibly due to increased LV filling (i.e., increased LVEDVI) in agreement with the Frank-Starling law, in addition to the systemic vasodilating effect of BPS which reduced the LV afterload, thus favouring the systolic antegrade flow.” Another question at regard is: are there other possible mechanisms by which this drug may influence positively the systolic function? Some specific cellular pathway related to calcium for example? Please, make a research also in experimental and in vitro models’ studies and expand if addition mechanisms have been demonstrated.
Response: Thank you very much for your suggestion. We have modified the sentences according to the Reviewer’s comment. As you mentioned, in addition to the vasodilating effect, the antiplatelet and anti-inflammatory effects of BPS may contribute to the reduction in PVI and consequently improve the pulmonary circulation and LV filling. We have added the following sentence into the discussion in dogs with PH.
Line 305-310: The improved LV function in dogs with pre-capillary PH was possibly due to increased LV filling (i.e., increased LVEDVI) in agreement with the Frank-Starling law, in addition to the systemic vasodilating effect of BPS which reduced the LV afterload, thus favouring the systolic antegrade flow. Furthermore, the antiplatelet and anti-inflammatory effects of BPS may also contribute to the reduction in PVI and consequently improve the pulmonary circulation and LV filling [12,13,39].
-line 275: typing error “BPS’“
Response: Thank you very much for your comment, and we deeply apologize the typing error. We have corrected the word.
Line 310-311: Overall, these results support BPS effectiveness for treating pre-capillary PH.
LIMITATIONS
I think that the list of limitations of the study should be expanded if Author will be not able to add the required lacking data. Moreover, an additional limitation may concerns the study designs. A better way to demonstrate the clinical efficacy of Beraprost respect the traditional approach would have been to design a case-control study where well-matched patients would receive Berparost vs Sildenafil (e.g., two 10-year, 10 kg mixed breed dogs treated for severe PH type due to a stage ACVIM C MMVD, one with Beraprost and one with Sildenafil).
Response: Thank you very much for your suggestions. As you mentioned, it is important to perform further studies comparing the clinical efficacy of BPS and sildenafil. We have added the following sentence into the limitation.
Line 336-338: Finally, we could not compare the efficacy of BPS and sildenafil. Further study comparing the clinical efficacy of BPS and sildenafil would be expected in the future.
Additional comments
In some part of the manuscript, you should say which is the cost for a daily treatment for a 10 kg dog treated with Beraprost at 15 μg /kg PO BID/kg, so that readers can compare the cost of Beraprost and Sildenafil and see which is the less expensive as Beraprost is new drug for veterinarians and many of us really ignore if its cost makes this drugs an available option in clinical practice or not.
Response: Thank you very much for your comment. In our institution, the delivered price is 1,242.5 Japanese yen per tablet for sildenafil and 66 Japanese yen per tablet for beraprost sodium. Therefore, in our institution, the price of 2.0 mg/kg sildenafil for a 10 kg dog would be 2.8 times higher than that of 15 µg/kg of beraprost sodium for a 10 kg dog. However, as for the selling price, we consider it difficult to state the price because of the possibility of errors from one facility to another.
